**Data Availability Statement:** Data has been provided as part of the submitted article.

**Funding:** This work was supported by grants from the British Columbia Immunization Committee

# What do adolescents think about vaccines? Systematic review of qualitative studies

Hana Mitchell [1], Rebecca Lim [1], Prubjot K. Gill [2], Joban Dhanoa [1], Ève Dubé [3], Julie A. Bettinger [1]*

**1** Vaccine Evaluation Center, BC Children's Hospital Research Institute, University of British Columbia, Vancouver, BC, Canada, **2** University of British Columbia Library, University of British Columbia, Vancouver, BC, Canada, **3** Institut National de Santé Publique du Québec, Québec City, Canada

* jbettinger@bcchr.ubc.ca

## Abstract

Adolescence presents a key opportunity to build vaccine-related health literacy and promote vaccine confidence and uptake. Although adolescents are central to vaccination programs, their views around vaccines are frequently underrepresented in qualitative literature. We reviewed qualitative studies to systematically identify and summarize existing evidence on adolescents' own understanding of vaccines and experiences with vaccine decision-making, including self-consent when applicable. CINAHL; Embase; Ovid Medline; and Psych Info database searches were last updated on May 28, 2022. Data pertaining to general study characteristics, participant demographics, and qualitative content were extracted independently by two reviewers and analyzed using textual narrative synthesis. Out of 3559 individual records, 59 studies were included. The majority of the studies were conducted in high-income countries and 75% focused on human papilloma virus vaccines, with the remaining studies looking at COVID-19, meningococcal, hepatitis B and influenza vaccines or adolescent experiences with vaccines in general. Adolescent self-consent was explored in 7 studies. Perspectives from sexual and gender minorities were lacking across studies. Adolescents often had limited understanding of different vaccines and commonly perceived vaccine information to be directed towards their parents rather than themselves. Many adolescents felt school-based vaccine education and information available through healthcare providers were insufficient to make informed decisions about vaccines. While adolescents described obtaining vaccine information from traditional and online media, face-to-face interactions and opinions from trusted adults remained important. Adolescents generally relied on their parents for vaccine-decision making, even when self-consent was an option. A notable exception to this included marginalized adolescents who could not rely on parents for health-related advice. Qualitative literature about adolescent vaccines would be enriched by studies examining vaccines other than the HPV vaccine, studies examining adolescent vaccine programs in low and middle-income countries, and by deliberately eliciting vaccine experiences of adolescent with diverse sexual orientation and gender identities.

(JAB), the Canadian Institutes of Health Research grant #151944 and the Public Health Agency of Canada grant #151944 (1 June 2017–31 December 2022) (JAB). The funders had no role in study design, data collection and analysis, decision to publish, or preparation of the manuscript.

**Competing interests:** The authors have declared that no competing interests exist.

## Background

Several vaccines are approved and recommended during adolescence, including human papilloma virus (HPV) series, meningococcal vaccine, hepatitis B, tetanus-diphtheria-acellular pertussis booster, influenza and most recently coronavirus-19 disease (COVID-19) [1–4]. Advances in vaccinology unfortunately have been accompanied by a global rise in vaccine hesitancy and refusal. Building and sustaining public confidence in vaccination programs among present and future generations is essential to mitigate a resurgence of vaccine preventable diseases [5–7]. Adolescence presents a key opportunity to promote healthy lifestyle behaviors including vaccination, thereby positively affecting future health choices [8–10].

Adolescents have the right to meaningfully participate in the design and delivery of interventions to improve and maintain their health [10, 11]. The importance of understanding and addressing adolescents' vaccine information needs and including them in vaccine decision-making is increasingly acknowledged [8, 10, 12–17]. Qualitative health research focuses on interpreting people's subjective experiences with healthcare and describing social contexts in which healthcare decisions occur. Systematically summarizing data obtained from qualitative studies is therefore useful to inform and guide development of appropriate and acceptable healthcare services for particular populations and to identify information needs and knowledge gaps [18–21]. Although adolescents are central to vaccination programs, their views around vaccines are frequently underrepresented in qualitative literature [8, 22] and warrant a dedicated systematic review.

The aim of this review was to systematically identify and summarize the existing qualitative evidence on adolescents' own understanding of vaccines and their experiences with participating in vaccine decision-making, including self-consent in jurisdictions where this is an option. This information is important to inform adolescent vaccination programs, to develop interventions to promote vaccine-related health literacy among adolescents, to support informed decision-making and to guide further research.

## Methods

### Protocol and eligibility criteria

The systematic review protocol was registered with Prospero (reference CRD42021267004) and is available in S1 Text. We included studies that provided primary qualitative data from adolescents published as full-text in a peer-reviewed journal or as a graduate thesis. Due to challenges with conducting detailed content analysis in multiple languages, only English language studies were considered. We did not place restrictions on publication date.

### Inclusion criteria

- Qualitative or mixed-methods studies.

- Studies included participants age 10 to 19, which is World Health Organization definition of adolescence [10]. Studies that also included data from younger or older individuals were included provided that majority of the participants were adolescents.

- Studies focused on adolescents' self-reported understanding of vaccines. We decided *a priori* to broadly apply the term understanding of vaccines. Depending on the study, understanding of vaccines could include a combination of adolescents describing their knowledge of vaccines and vaccine preventable diseases, their questions and concerns about vaccines, personal views and attitudes, as well as experiences with being involved in vaccine decision-making with or without legally self-consenting. In studies where adolescents had option to

self-consent, we looked at whether self-consent was implemented and whether adolescents perceived they had sufficient information to make a decision about being vaccinated.

- Qualitative data was obtained directly from adolescents and was reported in sufficiently rich detail to illustrate adolescents' self-described vaccine understanding or their involvement in vaccine decision-making–as deemed by the reviewers (HM, JD, MA, RL) and agreed through consensus.

## Exclusion criteria

- Studies where the qualitative component was deemed insufficient to contribute to further analysis (e.g. only brief illustrative quotes from adolescents without further context)

- Studies obtained qualitative data but reported it only quantitatively

- Qualitative studies focused strictly on evaluating specific educational interventions (e.g. evaluation of a video game about vaccines) or experiences with being vaccinated (e.g. managing pain)

## Information sources and search strategy

We searched CINAHL; Embase; Ovid Medline and Psych Info databases from inception to July 8, 2021 and then updated the search on May 28, 2022 to capture any additional studies. A comprehensive search strategy was developed for Ovid Medline and then modified for other databases with full strategies provided in S1 Table. All study titles and abstracts were imported into Covidence systematic review software, Veritas Health Innovation, Melbourne, Australia.

## Study selection and data collection

Study title and abstract screening and full text screening were done in Covidence with two out of four reviewers (HM, JD, MA, RL) independently assessing each study, using predetermined criteria. Disagreements were resolved by consensus.

Relevant data were extracted and entered into an Excel sheet in duplicate, independently by two reviewers (JD, RL), and then accuracy-checked and consolidated by a third reviewer (HM). We extracted data pertaining to general study characteristics and methodology, participant demographics, and qualitative content which is described in detail in the summary measures section below.

## Quality assessment

Quality was assessed using Critical Appraisal Skills Program (CASP) quality assessment tool [23]. Studies were included regardless of their methodological quality because insights from study participants were still considered to inform our synthesis [24], similar to the approach in other qualitative systematic reviews [19, 21, 25]. We discuss observed limitations of included studies as part of our findings.

## Summary measures and synthesis of results

Results were analysed using textual narrative synthesis approach as described by Lucas et al. [26] in which individual study characteristics and findings are reported according to a

standard format and structured summaries developed to elaborate on and putting into context the data extracted across studies [26, 27]. Categories used to report the results were developed in advance as per the study protocol. Qualitative content extracted included information about adolescents' self-reported understanding of vaccines and vaccine preventable diseases; knowledge gaps and misconceptions around vaccines; concerns and questions about vaccines or vaccination programs; vaccine information sources available to adolescents or additional information sources adolescents wished for; and adolescents' experiences with vaccine decision-making and self-consent as applicable. Qualitative data extraction was done by two reviewers (RL, JD) and entered into Excel spreadsheet. Following initial data extraction, a third reviewer (HM) again reviewed all the studies and consolidated the data extracted. Synthesis of findings from individual studies was done by (HM) with input from other study authors (RL, JD, ED, JAB), focusing on the scope, differences and similarities among studies to make broader observations about adolescents' understanding of vaccines and experiences with vaccine decision-making [26].

## Results

### Search results

Out of 3559 individual records identified, 59 studies were eligible for inclusion–Fig 1. Three studies in Australia [28–30], three in France [31–33] and two in the United Kingdom [34, 35] reported on data from the same set of interviews, but addressed distinct research questions and were therefore all included in the final review as individual studies.

### Study characteristics

Study characteristics are summarized in Table 1. Studies were conducted predominantly in high-income countries, with only 5 studies reported in low-to-middle income countries (LMICs). Individual study data was collected between 2003 and 2021. The majority of the studies (45/59) focused only on the HPV vaccine with remaining studies examining vaccines in general (6), COVID-19 (3), influenza (2), meningococcal B (1), meningococcal ACYW (1), and hepatitis B (1) vaccines. Vaccines were available to participants through a school-based immunization programs (SBIPs) in 22 studies. Most studies provided information on participants' vaccination status. Only 4 studies specifically reported how participants self-identify in terms of gender.

### Quality assessment

The overall quality assessment of included studies is summarized in Fig 2. The majority of the studies did not include any reflection on the relationship between researcher(s) and participants and how this may have affected study findings. Some studies did not provide enough information to assess whether data analysis was sufficiently rigorous. Table 2 provides an overview of each study.

### Study findings

**Adolescents' understanding of vaccines.** *Purpose of vaccines*. Study participants generally recognized the purpose of vaccines was protection from disease, although some adolescents were unclear on whether vaccines were preventative or therapeutic–as exemplified by studies examining understanding of influenza, HPV, or vaccines in general [16, 36–40]. For example, some middle and high school students believed the influenza vaccine can help cure the flu [39]. In another study, adolescent girls debated whether HPV vaccine should be given before

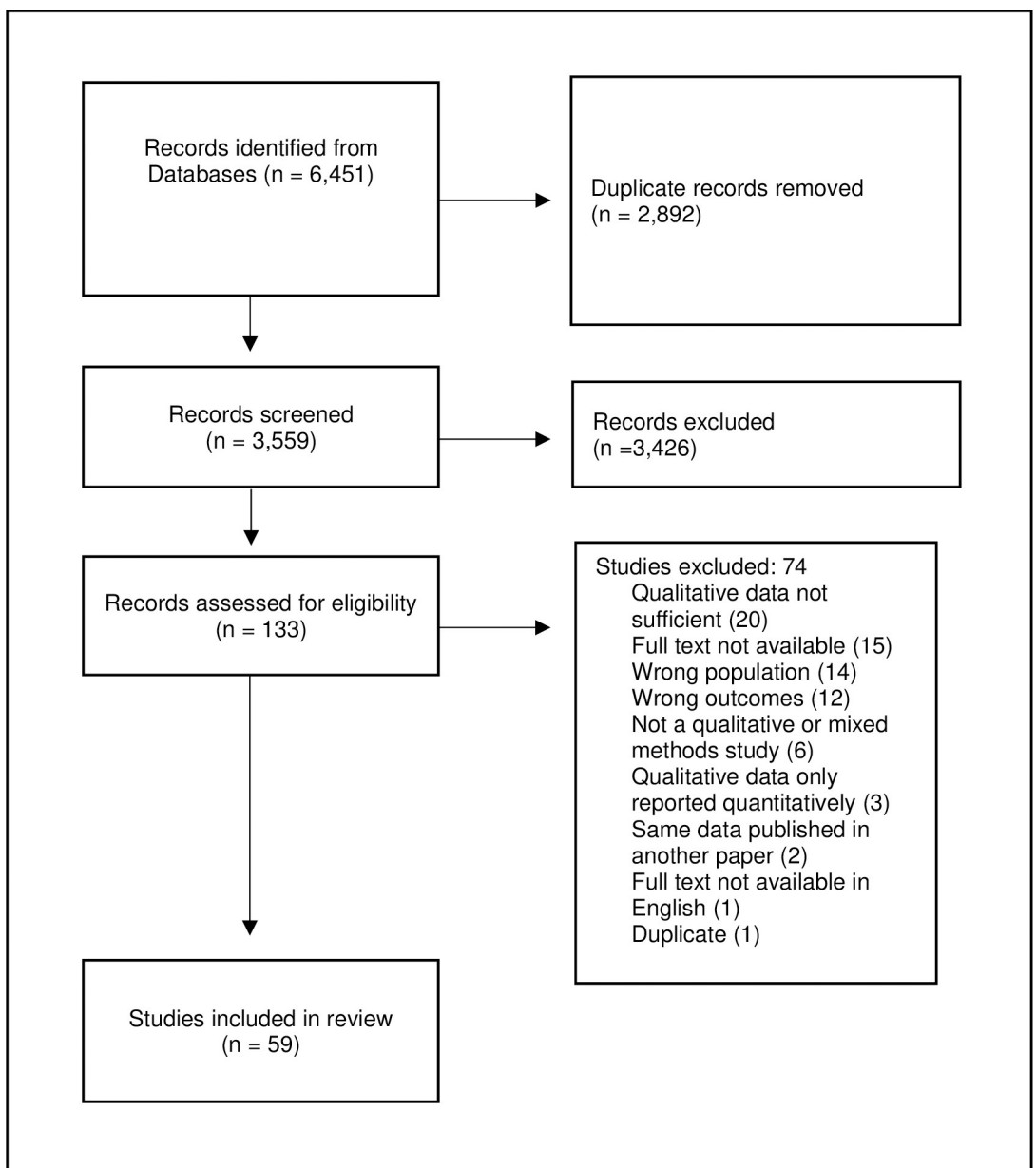

**Fig 1. PRISMA flowsheet.**

or after sex, reflecting confusion between prevention rendered by vaccination versus therapy after infection has potentially occurred [36]. Along with confusion between preventative versus therapeutic role of vaccines, a US study found that some adolescents also confused vaccines with other injectable substances such as contraceptive injection (depo-provera) and the tuberculin skin test [16].

While adolescents usually thought about vaccines in terms of personal protection, participants in a Swedish COVID-19 study and a UK study about various vaccines articulated the importance of being vaccinated to protect society at large [41, 42]. In 4 HPV studies conducted among girls (sexual orientation not reported) [31, 43], boys (sexual orientation not reported) [44], and boys who identified as either heterosexual or men who have sex with men (MSM)

**Table 1. Characteristics of included studies.**

| First author & Year of Publication | Study Aim | Country & Data collection period | Data collection method | Study included vaccine counselling/ education? | Number of participants by gender, sexual orientation (if reported) and age (years) | Participants' ethnicity* | Proportion of participants vaccinated | Vaccine(s) studied Participants have option to self-consent** | Vaccine(s) funded *** Vaccine(s) available through SBIP |
|---|---|---|---|---|---|---|---|---|---|
| Albert, 2019 | Examine the relationship between schools, families, parents, teachers, and girls as they interact with and make decisions around the HPV vaccine, health, and sexual health | Canada 2015–2016 | Semi-structured interviews | No | 19 girls All identified as heterosexual Age 11–17 | White, Chinese-Canadian, Lebanese, Guyanese | 9/19 | HPV No/ not stated | Yes Yes |
| Alexander et al., 2012 | Examine the decision-making process of parent-son dyads when deciding to get the HPV vaccine | USA Not specified | Semi-structured interviews | Yes, participants and parents spoke with HCP as part of vaccine counselling | 21 boys Age 13–17 | Black, Hispanic, White | 19/21 | HPV No/ not stated | Not specified No |
| Ali, 2021 | Explore adolescent girls' knowledge and perspectives on HPV and cervical cancer and collect their recommendations for implementing an HPV vaccination program in their community | Pakistan 2020 | Focus groups | No | 48 girls Age 16–18 | Muhajir, Punjabi, Sindhi, Pashto, Balochi | 0 | HPV No/ not stated | Not specified No (to be introduced) |
| Audrey et al., 2020 | Examine how acceptable new consent procedures are to young women, parent/carers, school staff and immunisation nurses | UK 2017–2019 | Semi-structured interviews | No | 19 girls Age 12–17 | 8 from ethnic minorities (not specified) | 19/19 | HPV Yes–although authors note it was rarely applied in practice | Yes Yes |
| Batista Ferrer et al., 2016 | Examine facilitators to uptake of HPV vaccine in an ethnically diverse group of young women | UK 2012–2013 | Semi-structured interviews | No | 23 girls Age 12–13 | White, Black, Asian, | 16/23 | HPV Yes–allowed under legal framework, however self-consent not applied by SBIP | Yes Yes |
| Benin, 2010 | Describe qualitatively how adolescents and parents understand and make decisions regarding the vaccine | USA 2003 | Semi-structured interviews | No | 25- gender not specified Age 10–14 | 16 African American 5 White 4 Hispanic | Not specified | General vaccines No/ not stated | Not specified No |

*(Continued)*

**Table 1.** (Continued)

| First author & Year of Publication | Study Aim | Country & Data collection period | Data collection method | Study included vaccine counselling/ education? | Number of participants by gender, sexual orientation (if reported) and age (years) | Participants' ethnicity* | Proportion of participants vaccinated | Vaccine(s) studied Participants have option to self-consent** | Vaccine(s) funded *** Vaccine(s) available through SBIP |
|---|---|---|---|---|---|---|---|---|---|
| Bernard et al., 2011 | Examine fear response and factors affecting fear in adolescents undergoing school-based HPV vaccination | Australia 2008–2009 | Focus groups, observation | No | 130 girls Age 12–16 | Not specified | Not specified | HPV No/ not stated | Yes Yes |
| Bland et al., 2009 | Examine factors influencing young adult decision making in regards to the Meningococcal vaccine | New Zealand 2007 | Semi-structured interviews | No | 1 boy, 10 girls Age 17–21 | New Zealand European/ European | 7/11 | Men B Yes | Yes Yes |
| Bond et al., 2016 | To better understand factors related to vaccine uptake among female adolescents from 3 racial groups | USA Not specified | Mixed methods: Focus groups & survey | No | 60 girls Age 13–19 | African American, Hispanic, Caucasian | 15/60 | HPV No/not stated | Not specified No |
| Boyd et al., 2018 | Determine the perceived barriers and facilitators to HPV vaccination among adolescents and their caregivers | USA 2014–2015 | Semi-structured interviews | No | 12 boys, 12 girls Age 11–18 | African American, Caucasian | 9/24 | HPV No/ not stated | Not specified No |
| Braunack-Mayer et al., 2015 | To investigate ethical issues in school-based immunization programs for adolescents and how they are addressed | Australia 2011 | Focus groups, semi-structured interviews | No | 38 girls Age 12–14 | Not specified | Not specified | General vaccines Allowed under legal framework, but not clear that option actually given in SBIP | Yes Yes |
| Budhwani, 2021 | Elucidate sentiments toward vaccination among African American or black adolescents | US | In-depth interviews | No | 15 girls 13 boys 15–17 | African American or Black | Not specified | COVID-19 No/not stated | Not specified No |
| Burns, 2021 | Examine the impact of knowledge and attitudes on HPV initiation and completion. | Australia 2016–2018 | Mixed Methods: Focus groups & survey | No | 70 -gender not specified Age 12–14 | Not specified | Not specified | HPV No/ not stated | Yes Yes |
| Chau, 2021 | Assess the HPV knowledge needs and program expectations of stakeholders involved in influencing the decision to accept HPV vaccination. | Hong Kong 2020 | Semi-structured interviews | No | 8 girls Age 12–17 | Chinese | Not specified | HPV No/not stated | Yes, grade 5–6 only No |

*(Continued)*

**Table 1.** (Continued)

| First author & Year of Publication | Study Aim | Country & Data collection period | Data collection method | Study included vaccine counselling/ education? | Number of participants by gender, sexual orientation (if reported) and age (years) | Participants' ethnicity* | Proportion of participants vaccinated | Vaccine(s) studied Participants have option to self-consent** | Vaccine(s) funded *** Vaccine(s) available through SBIP |
|---|---|---|---|---|---|---|---|---|---|
| Cooper Robbins et al., 2010 (a) | Explore and examine knowledge about HPV and HPV vaccine post-implementation of mass HPV vaccination in schools | Australia 2008–2009 | Focus groups | No | 130 girls Age 12–16 | Not specified | Not specified | HPV No/ not stated | Yes Yes |
| Cooper Robbins et al., 2010 (b) | Explore experiences, knowledge, attitudes, decision-making processes, and contextual factors related to consent to vaccination and vaccination completion in a school-based HPV vaccination program for adolescent girls | Australia 2008–2009 | Focus groups | No | 130 girls Age 12–16 | Not specified | Not specified | HPV No/ not stated | Yes Yes |
| Cordoba-Sanchez et al., 2019 | To identify barriers and facilitators of HPV vaccine uptake among girls eligible for vaccination and their parents in the initial years of vaccine implementation | Colombia 2016–2017 | Focus groups, semi-structured interviews | No | 49 girls Age 13–15 | Not specified | 38/49 | HPV No/not stated | Yes Yes |
| Dal Col Barthes, 2020 [116] | Explore the perception of cervical cancer vaccine among your girls | France 2016 | Semi-structured interviews | No | 34 girls Age 11–15 | Not specified | 10/34 | HPV No/ not stated | Not specified No |
| Doroshenko et al., 2012 | To explore challenges to obtaining essential vaccines experienced by homeless youth. | Canada 2009 | Focus groups | No | 16 boys, 1 girl Age 15–24 | Not specified | Not specified | General vaccines Yes–emancipated minors | Yes Yes– however homeless youth not in school |
| Fisher et al., 2020 | Consider the extent to which young women were able to exercise autonomy within the context of the HPV vaccination programme | UK 2018–2019 | Semi-structured interviews | No | 19 girls Age 12–17 | 8 from minority ethnic groups (not further specified) | 19/19 | HPV Allowed under legal framework, but self-consent not applied by SBIP | Yes Yes |

(*Continued*)

**Table 1.** (Continued)

| First author & Year of Publication | Study Aim | Country & Data collection period | Data collection method | Study included vaccine counselling/ education? | Number of participants by gender, sexual orientation (if reported) and age (years) | Participants' ethnicity* | Proportion of participants vaccinated | Vaccine(s) studied Participants have option to self-consent** | Vaccine(s) funded *** Vaccine(s) available through SBIP |
|---|---|---|---|---|---|---|---|---|---|
| Galbrath-Gyan, 2019 [117] | To increase understanding about the health beliefs of African-American parents and their daughters towards HPV infection and HPV vaccine acceptance. | USA 2014–2015 | Semi-structured interviews | No | 34 girls Age 12–17 | African-American | 12/34 | HPV No/ not stated | Not specified No |
| Garcia, 2021 | Explore vaccine perceptions of Latino families to inform culturally centered strategies for vaccine dissemination | USA 2020–2021 | Semi-structured interviews | No | 12 girls 7 boys 5 non-binary/ transgender/ non-conforming Age: average 16, grade 9–12 | Latino American | 0/24 | COVID-19 No/not stated | No Not specified |
| Getrich et al., 2014 | Examine vaccination decision making processes among clinicians, parents, and adolescents to identify strategies to enhance uptake | USA 2009–2010 | Mixed methods: focus groups & survey | No | 12 girls Age 12–17 | Hispanic, White | 6/22 | HPV No/ not stated | Not specified No |
| Gonzalez-Cano, 2021 | Understand how adolescent sexual behavior is approached in families, how widespread knowledge about HPV is, learn about the interviewees' position regarding vaccination | Spain 2017 | Focus groups | No | 55 boys, 82 girls Age 14–17 | Not specified | 61/137 | HPV No/not stated | No Yes |
| Gowda et al., 2012 | Identify similarities and differences between adolescents, parents and their HCPs in regards to their vaccination attitudes and practices | USA Not specified | Focus groups | Vaccine counselling by HCP | 10 boys, 22 girls Age 11–18 | Not specified | Not specified | General vaccines No/ not stated | Not specified No |

(*Continued*)

**Table 1.** (Continued)

| First author & Year of Publication | Study Aim | Country & Data collection period | Data collection method | Study included vaccine counselling/ education? | Number of participants by gender, sexual orientation (if reported) and age (years) | Participants' ethnicity* | Proportion of participants vaccinated | Vaccine(s) studied Participants have option to self-consent** | Vaccine(s) funded *** Vaccine(s) available through SBIP |
|---|---|---|---|---|---|---|---|---|---|
| Grandahl et al., 2019 | To explore awareness and thoughts about HPV and HPV vaccination, information sources, perceived benefits of vaccinating men, and intention to be vaccinated in a group of male upper secondary school students | Sweden 2017 | Semi-structured interviews | Nurse led education session at school for some of the participants | 31 boys Age 16–19 | Not specified | 0/31 | HPV Not stated, implied boys could self-consent | Yes (when available for males) No (to be introduced) |
| Griffioen et al., 2012 | To explore factors influencing mothers' decisions to vaccinate their 11–12-year-old daughters | USA Not specified | Semi-structured interviews | No | 33 girls Age 11–12 | Black White Multiracial | Not specified | HPV No/ not stated | Not specified No |
| Gutierrez et al., 2013 | To understand the perceptions of HPV and HPV vaccine among adolescent males | USA 2010 | Mixed methods: Questionnaire& focus groups | No | 86 Boys 41 identified as MSM Age: 13–21 (mean age 17) | Majority African American, White, Asian | 0/86 | HPV No/ not stated | Not specified No |
| Henderson et al., 2011 [118] | To examine the level of understanding and decision making among parents and girls regarding the HPV vaccine | UK 2008–2010 | Semi-structured interviews | No | 44 girls Age 12–15 | White | Not specified | HPV No/not stated | Yes Yes |
| Herbert et al., 2013 | Present the qualitative findings from parents and students who participated in a school-based influenza vaccination clinic | USA Not specified | Focus groups | Yes | 21 gender unspecified Age not specified (middle or high-school) | Majority African American | Not specified No/not stated | Influenza Not stated | Yes Yes |
| Hilton et al., 2011 | Identify gaps in adolescent girls' knowledge of HPV and its link to cervical cancer | UK 2009–2010 | Focus groups | No | 87 girls Age 12–18 | Not specified | 79/87 | HPV Yes, unclear whether actually applied in practice | Yes Yes |
| Hilton et al., 2013 | Explore teenagers' understandings, beliefs and experiences of diseases routinely vaccinated against | UK 2010–2011 | Focus groups | No | 30 boys, 29 girls Age 13–18 | Not specified | 46/59 | General vaccines Yes, unclear whether actually applied in practice | Yes Yes |

(Continued)

**Table 1.** (Continued)

| First author & Year of Publication | Study Aim | Country & Data collection period | Data collection method | Study included vaccine counselling/education? | Number of participants by gender, sexual orientation (if reported) and age (years) | Participants' ethnicity* | Proportion of participants vaccinated | Vaccine(s) studied Participants have option to self-consent** | Vaccine(s) funded *** Vaccine(s) available through SBIP |
|---|---|---|---|---|---|---|---|---|---|
| Hinds et al., 2004 | Investigate attitudes of school pupils and parents toward universal adolescent HepB vaccination | UK 2001 | Focus groups | Yes. Information sheet given to participant a week prior to focus group. | 20 boys, 30 girls Age 12–13 | Majority Caucasian | Not specified | Hep B No/ not stated | Yes No (to be introduced) |
| Hughes et al., 2011 | Generate hypotheses to inform interventions to increase HPV vaccine receipt | USA 2010 | Semi-structured interviews | No | 20 girls Age 11–12 | Black, White | 9/20 | HPV No/ not stated | Yes No |
| Hull et al., 2014 | To generate recommendations for framing messages to promote HPV vaccination for the undecided African American adolescents and their parents | USA Not specified | Focus groups, individual interviews | No | 21 girls Age 11–18 | African American | 9/21 | HPV No/ not stated | Not specified No |
| Karafillakis et al., 2021 | Explore the role of maturity in decision-making around HPV vaccination. | France 2018–2019 | Semi-structured interviews and focus groups | No | 36 girls Age 15–16 | Not specified | 9/36 | HPV No/not stated | Not stated No |
| Karafillakis et al., 2022 (a) | Explore the role of trust in HPV vaccination decision-making among mothers and adolescent girls. | France 2018–2019 | Semi-structured interviews and focus groups | No | 36 girls Age 15–16 | Not specified | 9/36 | HPV No/not stated | Not stated No |
| Karafillakis et al., 2022 (b) | An in-depth exploration and comparison of French mothers and adolescent girls' perceptions of the risks and benefits of HPV vaccination | France 2018–2019 | Semi-structured interviews and focus groups | No | 36 girls Age 15–16 | Not specified | 9/36 | HPV No/not stated | Not stated No |
| Katz et al., 2013 | Identify and compare barriers to HPV immunization perceived by healthcare providers, Black and Latino adolescents, and their caregivers | USA 2014 | Semi-structured interviews | No | 12 boys, 12 girls Age 12–17 | Black, Latino | 12/24 | HPV No/ not stated | Yes, for 80% No |

*(Continued)*

**Table 1.** (Continued)

| First author & Year of Publication | Study Aim | Country & Data collection period | Data collection method | Study included vaccine counselling/ education? | Number of participants by gender, sexual orientation (if reported) and age (years) | Participants' ethnicity* | Proportion of participants vaccinated | Vaccine(s) studied Participants have option to self-consent** | Vaccine(s) funded *** Vaccine(s) available through SBIP |
|---|---|---|---|---|---|---|---|---|---|
| Katz et al., 2013 | To elucidate factors influencing HPV vaccination low-income South African adolescents receiving the vaccine for the first time in Soweto | South Africa Not specified | Semi-structured interviews | No | 16 boys, 23 girls Age 12–19 | Black African | Not specified | HPV No, but parental consent perceived as "formality"– decision by adolescents | Yes No |
| Kemberling et al., 2011 | Understand the knowledge levels, attitudes and perceptions of Alaska Native adolescent girls about cervical cancer, HPV, genital warts and the HPV vaccine | USA 2008 | Mixed methods: semi-structured interviews & survey | Yes | 79 girls Age 11–18 | Native American | Not specified | HPV No/ not stated | Not specified No |
| Kwan et al., 2008 | Explore perceptions towards cervical cancer, HPV infection and HPV vaccination and to identify factors affecting the acceptability of HPV vaccination among Chinese adolescent girls in Hong Kong | Hong Kong 2007 | Mixed methods: focus groups, surveys | No | 64 girls Age 13–20 | Chinese | Not specified | HPV No/ not stated | No No |
| Lefevre et al., 2019 | Explore experiences and representations of HPV vaccination by adolescent girls seeing doctors at least occasionally | France Not specified | Written essays (96), semi-structured interviews (5) | No | 101 girls Age 11–19 | Not specified | 14/101 | HPV No–authors specify parental consent required | No No |
| Marshall et al., 2019 | Identify factors that influence the adolescent HPV vaccine decision and identify strategies to reduce vaccine hesitancy | Ireland 2017–2018 | Focus groups | No | 50 girls Age 14–16 | Not specified | 48/50 | HPV No/ not stated | Yes No |
| Miller et al., 2014 [119] | Explore attitudes and beliefs about HPV vaccination and identify perceived barriers to HPV vaccination among urban, minority, economically disadvantaged adolescents | USA 2012 | Mixed methods: focus groups & survey | No | 18 boys 32 girls 2 girls identified as bi-sexual, other participants as heterosexual Age 14–18 | Primarily Black | 20/50 | HPV No/ not stated | Not specified No |

*(Continued)*

**Table 1.** (Continued)

| First author & Year of Publication | Study Aim | Country & Data collection period | Data collection method | Study included vaccine counselling/ education? | Number of participants by gender, sexual orientation (if reported) and age (years) | Participants' ethnicity* | Proportion of participants vaccinated | Vaccine(s) studied Participants have option to self-consent** | Vaccine(s) funded *** Vaccine(s) available through SBIP |
|---|---|---|---|---|---|---|---|---|---|
| Morales-Campos et al., 2013 [120] | Assess Hispanic mothers' and girls' perceptions about cervical cancer, HPV, and the HPV vaccine | USA 2008–2009 | Focus groups | No | 28 girls Age 14–18 | Hispanic | 8/28 | HPV No/ not stated | Not specified No |
| Nilsson, 2021 | Explore Swedish adolescents' willingness to be vaccinated against COVID-19 | Sweden 2020 | Survey with qualitative component | No | 296 boys, 404 girls Age 15–19 | Not specified | No | COVID-19 No/not stated | Not specified Not specified |
| Nodulman et al., 2015 | Investigate the attitudes and knowledge of stakeholders regarding the HPV and HPV vaccine | USA 2010 | In-depth interviews and focus groups | No | 38 girls Age 11–12 | African American, American Indian, Hispanic, non-Hispanic White | Not specified | HPV No/not stated | Not specified No |
| Occa et al., 2020 | Identify gaps and misperceptions about vaccines and provide recommendations to develop educational interventions | Italy Not specified | Focus groups | No | 24 boys, 30 girls Age 11–14 | Italian | Not specified | HPV No/ not stated | Yes Yes |
| Oostdijk, 2021 | Examine MenACWY decision-making process and dynamics within households, taking perspectives of both parents and adolescents into account | Netherlands 2019 | Semi-structured interviews | No | 6 boys, 12 girls Age 14–18 | 4/18 non-Western | 13/18 | Men ACWY Yes | Yes No |
| Ramanadhan et al., 2020 | Explore the attitudes of adolescents about using community setting to deliver the HPV vaccine | USA 2018 | Focus groups | No | 8 boys, 14 girls Age 11–14 | Latino, Black, White, Asian/Pacific Islander, Brazilian | 0/22 | HPV No/ not stated | Not specified No |
| Short et al., 2014 | Describe adolescents' perspectives regarding the use of school-located immunization programs for Influenza vaccination | USA Not specified | Focus groups | No | 30 boys, 25 girls Middle or high school | Hispanic, Black, White, Asian, Pacific Islander | Not specified | Influenza No/ not stated | Yes Yes |

(*Continued*)

**Table 1.** (Continued)

| First author & Year of Publication | Study Aim | Country & Data collection period | Data collection method | Study included vaccine counselling/ education? | Number of participants by gender, sexual orientation (if reported) and age (years) | Participants' ethnicity* | Proportion of participants vaccinated | Vaccine(s) studied Participants have option to self-consent** | Vaccine(s) funded *** Vaccine(s) available through SBIP |
|---|---|---|---|---|---|---|---|---|---|
| Turiho et al., 2017 | Perceptions of the HPV vaccination and their perceived implications for acceptability of HPV vaccination of adolescent girls | Uganda 2011 | Mixed methods: Focus groups & survey | No | 43 girls Age 13–16 | Ugandan | 43/43 | HPV No/ not stated | Yes Yes |
| Vardeman et al., 2008 | Explore how teen girls and parents of teen girls understand HPV vaccine communication campaign. | USA Not specified | Semi-structured interviews and focus groups | No | 1 boy, 39 girls Age 13–18 | White, Black, Latina, Biracial | Not specified | HPV No/ not stated | Not specified No |
| Virtanen and Salmivaara, 2021 | Ask how HPV vaccine is framed in the daily lives of vaccination- aged Finnish girls and in school nurses' everyday work | Finland, Not specified | Semi-structured interviews | No | 12 girls Age 10–12 | Not specified | 12/12 | HPV Yes | Yes Yes |
| Wakimizu et al., 2015 | Examine HPV vaccination process for Japanese adolescent girls and factors influencing vaccine decisions | Japan 2011–2012 | Semi-structured interviews | No | 20 girls Age 12–17 | Japanese | 16/20 | HPV No/ not stated | Funded -for girls age 13–16 No |
| Williams et al., 2011 | Explore knowledge about HPV and HPV vaccine attitudes among girls who were part of vaccine "catch up" program | UK 2009 | Semi-structured interviews | No | 10 girls Age 17–18 | White, Asian | 7/10, 1 girl only received 1 dose | HPV Yes | Yes Yes, but catch up through GP |
| Zeraiq et al., 2015 | Gain insight into Arabic mothers' and daughters' attitudes towards the HPV vaccine, and the reasons and conditions for accepting or rejecting it | Denmark 2011–2012 | Focus groups | No | 13 girls Age 12–18 | Middle Eastern | Not specified | HPV No/not stated | Yes Yes |

(*Continued*)

**Table 1.** (Continued)

| First author & Year of Publication | Study Aim | Country & Data collection period | Data collection method | Study included vaccine counselling/ education? | Number of participants by gender, sexual orientation (if reported) and age (years) | Participants' ethnicity* | Proportion of participants vaccinated | Vaccine(s) studied Participants have option to self-consent** | Vaccine(s) funded *** Vaccine(s) available through SBIP |
|---|---|---|---|---|---|---|---|---|---|
| Zipursky et al., 2010 | Examine knowledge and attitudes towards vaccines and immunization amongst adolescents in South Africa | South Africa 2008 | Focus groups | No | 32 boys, 31 girls Age 15–18 | Not specified | Not specified | General vaccines Not clear, legal framework not established | Some vaccines funded by government No |

GP–General practitioner

HCP–Healthcare provider

Hep B—Hepatitis B

HPV–Human papilloma virus

Men ACWY–Quadrivalent meningococcal vaccine

Men B–Meningococcal B vaccine

MSM–Men who have sex with men

SBIP–School based immunization program

*Participant ethnicity–as defined by study authors

** Option to self-consent–as described in the context of the study, not necessarily related to jurisdiction's legal framework

*** Vaccine funded for participants at time of the study through public funding or insurance program

[45] respectively, participants articulated the importance of HPV vaccination to protect their partner.

*Risk of vaccine preventable diseases.* Studies found adolescents often had difficulty understanding their present and future disease risk. Adolescent participants frequently did not perceive themselves to be at risk for infection and voiced uncertainty about why specific vaccine (s) were recommended for them. In some cases, this reflected participants' unfamiliarity with disease transmission. For example, in a study focusing on hepatitis B vaccine, some adolescent participants did not understand how hepatitis B virus was transmitted [46]. In contrast, a New Zealand study examining meningococcal B vaccine showed that while older adolescents understood how meningitis was transmitted, many found it difficult to conceptualize meningitis infection as something that could happen to them [47]. Some adolescents felt COVID-19 was not "their disease" because older age is a known risk factor for severe infection [42].

Several HPV studies conducted in high as well as LMIC settings highlighted adolescents confused HPV with other sexually transmitted infections, such as herpes, human immunodeficiency virus, or gonorrhea, further contributing to misunderstandings about their personal risk of acquiring HPV and the purpose of HPV vaccines [28, 40, 43, 48–51].

*Concerns about vaccines.* Adolescents were commonly concerned about pain with vaccines–a worry observed across all vaccines being studied. Some adolescents receiving vaccines through SBIPs also voiced concerns about lack of privacy during vaccination, including influenza [39], HPV [52, 53] and vaccines in general [41]. In five studies conducted in France (HPV) [32] South Africa (vaccines in general) [40], United Kingdom (HPV) [53] and United States (influenza, vaccines in general) [16, 39] adolescents were concerned about needle safety and questioned whether school or community clinics provided a sanitary setting for delivery of vaccines.

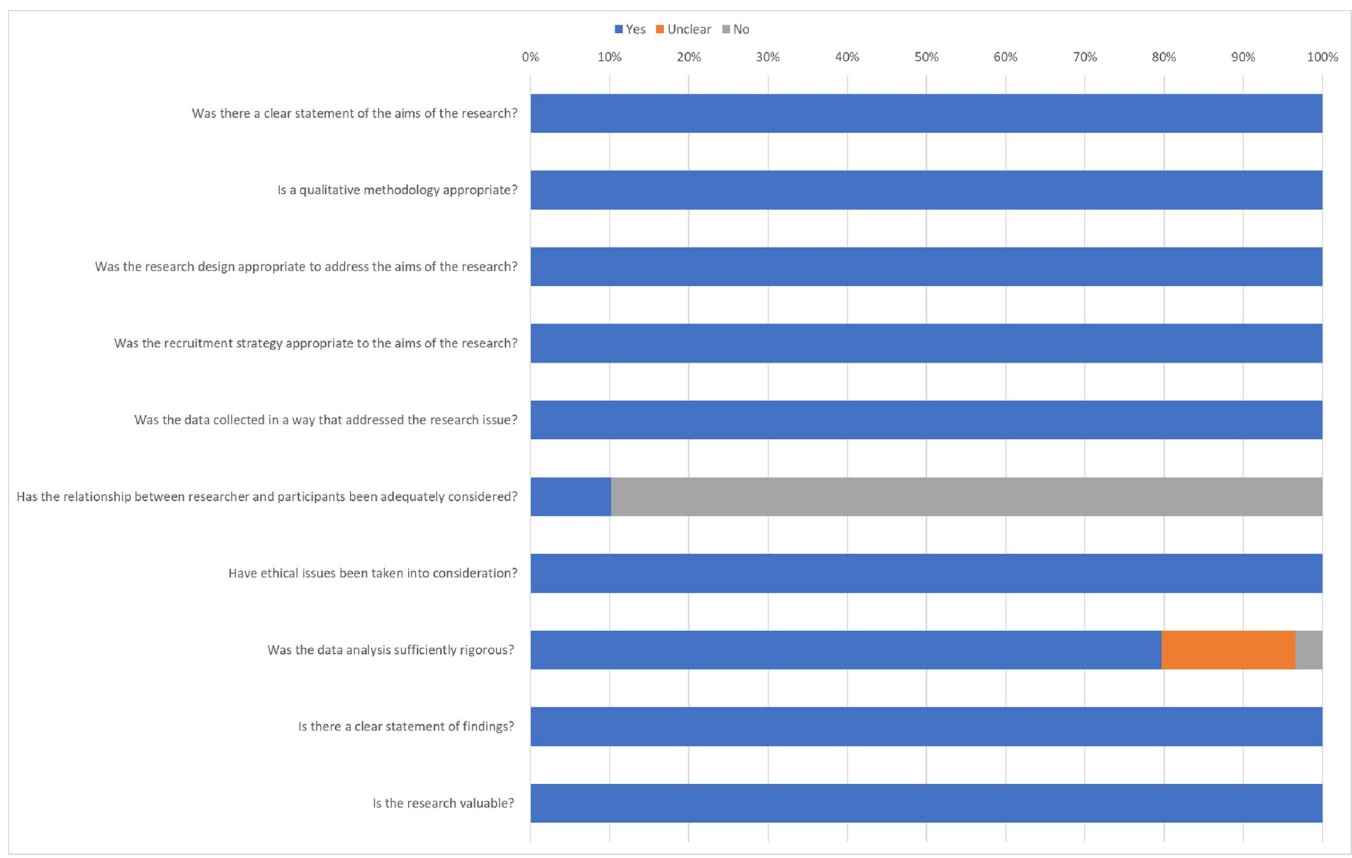

**Fig 2. Critical Appraisal Skills Program CASP evaluation summary for included studies.**

Specific fears around long-term side effects of vaccines were highlighted in HPV studies, where some adolescents worried HPV vaccine caused paralysis [43, 50, 54, 55], impaired the body's natural immunity to fight infections [30, 53], affected physical appearance [36, 50], affected fertility [35, 36, 41, 50, 55], led to unknown long term side-effects [45, 47, 56, 57], or caused the HPV infection or cancer [53, 57]. Sources of (mis)information about HPV vaccine and its side effects were not explored in detail in any of the studies but appeared to have originated primarily from parents or other care takers and less commonly peers or the media.

Studies looking at vaccines other than HPV did not describe specific worries or questions around long term side effects, although it was not clear whether this was because adolescents were not concerned about specific side effects or because studies did not elicit such information. Two US COVID-19 studies reported adolescents thought vaccine program was part of government conspiracy [58, 59], a sentiment not conveyed in any of the studies about other vaccines.

**Vaccine information sources described by adolescents.** *Parents or caretakers.* Parents or other caretakers were consistently described as the main source of vaccine information by adolescents of all ages. Because parents were usually required to provide consent for and sometimes pay for the vaccine, adolescents in many studies explicitly or implicitly assumed parents had or were responsible for obtaining and evaluating vaccine information. For example, vaccine information sheets received through SBIPs were commonly perceived to be intended for parents rather than adolescents. Similarly, adolescents usually felt that information provided by healthcare providers was directed towards parents even when adolescents were present for

**Table 2. Summary of qualitative findings.**

| Adolescents' understanding of vaccines | Vaccine information sources described by adolescents | Vaccine decision-making |
|---|---|---|
| **Purpose of vaccines:**<br>• Understand general purpose, but often unclear about specific vaccines<br>• May be unsure whether vaccines preventative or therapeutic<br>• Tend to think about vaccines in terms of personal rather than public protection | **Parents or caretakers:**<br>• Usually main source of information<br>• Generally perceived by adolescents as being responsible for obtaining and evaluating vaccine information<br>• Vaccines usually discussed at home at the time of vaccine consent | **Lead by parents or caretakers:**<br>• Parents generally required to provide vaccine consent<br>• Adolescents may have pre-formulated opinions about vaccines, but final decision is generally parental, even when adolescent self-consent is an option |
| **Vaccine preventable diseases:**<br>• Difficulties understanding and differentiating present and future risk<br>• May be unfamiliar with diseases and disease transmission | **Schools:**<br>• Many adolescents wish for more vaccine information through schools<br>• Vaccine information sheets not sufficient, adolescents wish for interactive sessions | **Lead by adolescents:**<br>• Self-consent often not offered in practice<br>• Sub-groups of adolescents from marginalized and minority groups may exercise higher degree of autonomy when it comes to vaccine decisions |
| **Concerns about vaccines:**<br>• Pain with vaccination<br>• Lack of privacy during vaccination<br>• Short and long-term side effects<br>• Questionable effectiveness | **Healthcare providers:**<br>• Trusted information resource, but not all providers initiate vaccine conversations<br>• Adolescents perceive providers address parents rather than adolescents when discussing vaccines | |
| | **Peers:**<br>• Not commonly described as key information source<br>• Can be source of support during school vaccine clinics | |
| | **Media**<br>• Increasing use of internet and social media<br>• Unclear how adolescents assess and triangulate media information<br>• In-person communication remains important | |

the appointments (see the sections on schools and healthcare providers below). With COVID-19 vaccine, some adolescents considered whether their parents or other family got vaccinated, since older adults became eligible for vaccination before adolescents and were therefore seen as role models [59].

Situations where adolescents did not rely on parents for vaccine information included adolescents whose parents did not speak the language used in healthcare communication or had low literacy levels, prompting adolescents to act as knowledge brokers to discuss vaccines with their parents [25, 40, 48, 49, 58, 60]. Some adolescents were uncomfortable talking about HPV vaccine with their parents because it would involve talking about adolescent's sexuality [36, 55, 61] and wished for adolescent friendly information. MSM participants described obtaining their own information regarding sexual health including HPV vaccine, especially when their family did not approve of their sexual orientation [45]. Homeless youth relied on healthcare information provided through youth shelters or through healthcare providers [37].

*Schools*. HPV vaccine studies generally reported that adolescents thought schools were an appropriate place to learn about the vaccine. Adolescents wished for more detailed vaccine information to be provided in school settings by healthcare professionals at the school and occasionally by teachers. This was true for adolescents in Australia [28, 30, 52, 62], Colombia [54], Denmark [60], Sweden [44] and Uganda [50] where HPV vaccine was available through SBIPs as well as for adolescents in several studies in the US [63–66], Ireland [67] and France [55] where HPV was not part of SBIPs.

In three studies [35, 44, 68] participating adolescents expressed face-to-face interactions with the opportunity to ask questions were preferred over receiving written or digital information. In another SBIP HPV study, researchers observed adolescents were often curious about

the science behind vaccines and vaccine preventable diseases—topics that could enrich classroom discussions [38].

Adolescents involved in SBIPs that only provided written information sheets about HPV vaccine, but no dedicated education sessions or opportunities to ask questions, often commented they did not feel well informed. For example, Australian adolescents remarked HPV vaccine information sheets they received in school were too complex and perceived they were written for parents rather than teenagers [29, 30, 52]. However, none of these studies elaborated in detail on what kind of information these adolescents wished to receive through schools.

Schools were also described as a trustworthy source of information among a handful of studies examining vaccines other than HPV. A study done in South Africa described school-going adolescents usually perceived healthcare providers and schools as credible resources, while parents were sometimes perceived as being less knowledgeable about vaccines because they were generally less educated [40]. US adolescents participating in a pilot project that provided classroom-based influenza vaccine education thought favourably of the program and agreed school was an appropriate venue to provide such information [68].

Finally, a Canadian study among homeless youth and a South African study that included non school-going adolescents highlighted how vaccine information provided through schools will not reach all adolescents and underscored the importance of identifying alternative information sources for these youth [37, 40].

*Community healthcare providers.* Community healthcare providers, such as family physicians or community nurses, were frequently mentioned by adolescents as trusted source of vaccine information. Several US studies where HPV vaccine is usually provided through physician's clinics examined adolescents' experiences with vaccine counselling done alongside with their parents. While adolescents appreciated being part of the discussion, they generally felt that vaccine information presented by their healthcare provider was directed primarily towards their parents, even when healthcare providers included adolescents in the conversation [31, 33, 49, 63, 69, 70]. It is worth noting that none of the studies explored vaccine counselling during confidential adolescent vaccine visits.

*Peers.* Peers were not commonly described as a source of vaccine information. Some adolescents said peers could be sources of rumours and fearmongering about vaccines, especially for vaccination pain. Vaccinated adolescents generally felt hearing about peers being vaccinated or having their friends accompany them when vaccinated at school made them more comfortable with being vaccinated as described in some of the HPV [29, 36, 48, 67, 71], influenza [39] and hepatitis B, studies [46]. Japanese, South African and UK adolescents who reported discussing vaccine with parents, indicated hearing about peers being vaccinated motivated them to being vaccinated [48, 71, 72].

*Media.* Many adolescents mentioned hearing about vaccines on the radio, television or on the internet. Social media was specifically mentioned in five studies (HPV, meningococcal vaccine) with four of these studies published in 2020 or later [33, 63, 73–75].

Studies that described radio, television, Internet or social media as information sources did not elaborate on what kind of information adolescents reported accessing or how they triangulated it with information from their parents, healthcare providers, or schools.

Assessing reliability of the information available through traditional and digital media was not explored in great detail. A 2019 study in Ireland emphasized that while adolescents acknowledged information available on the internet was not always trustworthy, they were not aware of specific websites where they could access reliable information [67].

A US study showed some girls preferred seeing HPV vaccine information on the television rather than receiving information at their school, because they often watched TV together

with their mothers which made it easier to bring up topic [73] highlighting the importance of providing vaccine information in a way that engages both parents and adolescents. In contrast, girls in a Pakistani study felt that television would not be an appropriate way to provide HPV information because it was mainly watched by men rather than women [74].

Several studies highlighted how adolescents were influenced by the current events popularized in the media. Among Colombian adolescents, HPV vaccine perceptions were negatively affected by local media reports featuring a young women who allegedly experienced debilitating side effects from the vaccine [54]. In contrast, some UK adolescents had heightened awareness of cervical cancer and HPV vaccine after young reality television star passed away from the disease [53] and French adolescent girls believed felt influencers could have a strong impact on vaccine perceptions by sharing personal experiences [33]. Media messages also influenced adolescents' perception of their disease risk. A New Zealand media story about a young infant who contracted meningitis B motivated some adolescents to be vaccinated, while other perceived only young children were at risk of infection [47].

**Decision making about vaccines.** *Adolescents who are able to provide self-consent.* Seven of the included studies specified that participating adolescents had the legal ability to consent to or decline vaccines under their local jurisdiction framework and examined whether self-consent was actually exercised as part of the vaccination program. This included five HPV studies in the UK (ages 12–18) [25, 34, 35, 72] and Finland (ages 10–12) [76]; two meningococcal vaccine studies conducted in the Netherlands (ages 14–17) [75] and New Zealand (ages 17–21) [47], as well as a general vaccine study among Canadian homeless youth (with those younger than 18 considered emancipated minors) [37]. In addition, two South African studies noted while the country's legal framework around adolescent vaccine consent was unclear, adolescents commonly had de-facto ability to consent with many adolescents growing up parentless, having parents who were overburdened with other tasks or parents who did not have sufficient literacy skills to interpret vaccine information [40, 48].

It is worth noting that while self-consent was available legally, the option to self-consent was not consistently offered to adolescents in real life. In three out of four UK HPV studies looking at HPV vaccine delivery through SBIPs, study authors observed self-consent was rarely applied [34] and noted SBIP staff were uncomfortable with providing self-consent option to adolescents [25, 35]. Similarly, nurses vaccinating girls through SBIP in Finland were concerned that vaccinating a girl against parental wishes may compromise the relationship between the nurse, the girl and the parent [76]. Self-consent was only consistently applied in one UK study which only included older adolescent women, ages 17–18 years [72]. In the Netherlands the consent form was sent to the household and it was up to the parents to decide whether they left the decision to be vaccinated with their teenager or not [75].

Among adolescents in high-income countries who self-consented, many mentioned that the vaccine information provided to them through public health was insufficient or that they did not perceive it to be completely reliable and free of bias. They reported talking to parents, teachers, peers and doing their own research on the Internet to gather enough information to make the decision [47, 72, 75]. Parents were described as a major influence several times for meningococcal vaccines, with adolescents ultimately following parental advice even when their opinions diverged and adolescents had the ability to provide self-consent [47, 75]. In contrast, adolescents in South Africa and homeless youth in Canada described making vaccine decisions primarily based on the information provided in schools or youth shelters and through healthcare providers, highlighting the importance of making vaccine information accessible to vulnerable adolescents who cannot rely on parents for vaccine decision making.

**Adolescents who require parental consent.** Several studies explored adolescent involvement in the vaccine decision-making process when the final decision and consent were

parental. While not all studies explicitly stated parental consent was required, this was usually implied in the description of interactions between adolescents, adults and other stakeholders. In settings where vaccines were not publicly funded adolescents relied on parents not only for consent, but also to cover the cost of the vaccine [36, 55, 71], further consolidating adolescents' perception that decision to be vaccinated was parental.

While some adolescents perceived being involved in vaccine discussions as a step towards autonomy [49, 77], others felt they were given a "false choice" in which the only legitimate option was to agree [41, 78] or simply deferred to parents to make the decision for them [28, 31, 33]. Several studies in both higher income and LMIC settings highlighted that providing vaccine information directly to adolescents may prompt discussions with their parents and potentially lead to better vaccine awareness and uptake [36, 40, 44, 48, 50, 68, 71]. For example, HPV vaccine campaigns in Hong Kong and Japan were targeted towards adolescent women who subsequently brought up the topic at home [36, 71]. Similarly, US adolescents participating in school-based influenza vaccination reported discussing what they learned in school with their parents [68]. In Uganda HPV vaccine for adolescent girls was promoted through the media, community outreach activities by healthcare workers and school-based vaccine education, resulting in high degree of acceptance among adolescent girls and their parents [50].

## Discussion

This systematic review included 59 qualitative studies on adolescents' own understanding of vaccines, vaccine information sources, and experiences with vaccine decision-making. With the vast majority of the studies conducted in high-income countries and three-quarters of the studies focused on HPV vaccines, we are able to make several observations about adolescents' attitudes towards vaccines in general, while highlighting important areas for further research where information is currently lacking.

Studies included in our review demonstrate adolescents have some understanding of vaccines and vaccine preventable diseases but are oftentimes unsure why a specific vaccine is recommended for them and may have questions and concerns about vaccine safety and effectiveness. Importantly, adolescents may perceive vaccine-related information provided in schools or via community healthcare providers as being directed towards their parents/caretakers and may not be motivated to seek out vaccine information for themselves or to actively participate in vaccine decision making.

While an incomplete understanding of vaccines or vaccine preventable diseases is certainly not unique to adolescents [21, 22, 79] it is important to acknowledge that adolescents may have different healthcare information needs and concerns than adults. They may also have different risk perceptions compared to adults and may be less motivated to seek preventative care [8].

A recent systematic review of primarily quantitative studies found that concerns about vaccine side effects and the lack of detailed vaccine information were two of the main reasons of hesitancy among adolescents [80].

Identifying pertinent vaccine information and appropriate communication methods to actively engage adolescents is important for increasing vaccine-related literacy and acceptability of vaccines among adolescents [17, 81–83]. Studies included in this systematic review show such information should be clearly intended for and communicated directly to adolescents. While adolescents may perceive their parents to be reliable sources of vaccine information and advice, parents themselves often may not be knowledgeable about vaccines [79, 84, 85], underscoring the importance of continuing to provide vaccine information to parents as well as adolescents.

When discussing vaccine information available to adolescents and vaccine decision making, it is important to acknowledge legal and socio-cultural contexts in which adolescent vaccination programs are taking place. The majority of the studies included in this review were done in settings where parental consent was required–as stated by the study authors or implied from the study context. Inability to provide self-consent is a barrier to vaccination during health care visits when parents are absent [12, 86]. In addition, vaccine cost may prevent adolescents from making independent decisions about vaccines in jurisdictions where vaccines are not publicly funded [22]. Studies included in this review were conducted before COVID-19 vaccine became available to adolescents. Subsequent roll-out of COVID-19 vaccine for adolescents in some higher income countries has highlighted parental as well as legislative tensions that can occur when young people wish to be vaccinated against parental wishes [87–89]–a topic that warrants further attention when it comes to routine adolescent vaccines.

Along with parents, schools and healthcare providers were described as reliable and trustworthy sources of vaccine information. Schools have the potential to play a vital role in providing adolescent-focused vaccine education [82, 90–93]. Our review confirmed adolescents generally consider schools to be a trustworthy place to learn about vaccines. While a number of countries have adopted SBIPs to facilitate vaccination access, our review demonstrated SBIPs are not necessarily accompanied by vaccine-related education beyond the information provided in vaccine information sheets, and may not provide opportunities for adolescents to ask questions about vaccines. While studies of SBIPs in this review were limited to HPV and meningococcal vaccines, we believe that this observation applies to adolescent vaccines more broadly.

SBIPs present an opportunity to not only facilitate vaccine delivery, but also engage adolescents in learning about vaccines and improve their vaccine knowledge and, by extension, potentially improve knowledge among their parents. For example, a recent randomized trial conducted in Australia demonstrated that adolescents who received HPV vaccine education as part of their SBIP had greater vaccine understanding and confidence and were more engaged with discussing the vaccine with their parents [83]. For jurisdictions without school SBIPs, schools may still be an important source of vaccine information for students, their parents and their communities [82, 90].

Unequivocal recommendation by a trusted healthcare provider is a well-recognized determinant of vaccine confidence and uptake among parents [22, 94–98] and may influence vaccine confidence and uptake among adolescents [99, 100]. Discussing vaccines presents an opportunity to support adolescents in making independent healthcare related decisions [8, 16], but requires careful navigation when parental consent is required, or when adolescent and their parents may have divergent views around vaccines. Studies included in our systematic review were limited to healthcare provider interactions that included both parents and adolescents, rather than confidential adolescent visits. A recent systematic review of healthcare provider strategies to increase HPV vaccine uptake found the use of multi-settings to target hard-to-reach vulnerable adolescents, consistently recommending the vaccine to adolescents, open-communication and motivational approaches, and sexual health education were some of the effective strategies used [100]. Studies exploring adolescents' own experiences with vaccine counselling in healthcare settings would be useful.

Even though traditional and online media were both described as vaccine information sources, adolescents looked to parents, healthcare professionals, and school staff for vaccine information and valued personal communication. This is in keeping with a recent literature review which found that even though many young people use digital health technologies they also found online information difficult to navigate and appreciated face-to-face interactions [101]. Global internet access as well as social media use has increased exponentially over the

last two decades, with the majority of adolescents now using social media [102]. It is important to recognize adolescents are increasingly exposed to and deliberately seeking healthcare information online. Empowering adolescents to learn and make informed decisions about vaccines cannot be separated from discussions about how to look for credible sources of digital health information [103–105]. Some of the studies included in this review were published when the Internet was less widely available and social media applications were just beginning to gain traction. Future studies examining adolescents' understanding of vaccines should include a more nuanced inquiry into how they access and interpret vaccine information available through digital and traditional media and how they triangulate this information with what they hear from their parents, healthcare providers, peers or at school.

Issues around vaccine information needs and acceptance among adolescents may differ depending on the specific vaccine. While there has been a number of studies published with regards to HPV vaccine, qualitative data about other vaccines recommended in adolescence are relatively limited. For example, meningococcal vaccine was introduced into adolescent vaccination program around the same time as HPV in many jurisdictions, but received limited attention in research studies. Given that suboptimal vaccine uptake during adolescence is by no means limited to HPV [90, 106–108], future studies should examine other vaccines routinely recommended during adolescents. In addition, more qualitative data from adolescents in LMICs would be helpful with accounting for diverse settings in which adolescent vaccines are being administered and informing vaccine education strategies.

Finally, perceived risk of infections, healthcare decision-making, and vaccination uptake may all be influenced by one's sexual orientation and gender identity [109–111]. People from sexual and gender minorities may have difficulties accessing and receiving healthcare. For example, a recent United States study found youth who identified as transgender or gender non-conforming reported lower rates of preventive health checkups [110]. Gender and sexual orientation diverse youths may also be more likely to seek and be exposed to health information online rather than from healthcare professionals, compared to heterosexual peers [112]. Understanding whether and how health-seeking behaviour and access is affected by sexual orientation and gender identity is therefore important for ensuring equitable access to vaccine information and vaccination. HPV vaccination programs, in particular, would benefit by accounting for sexual orientation and gender identify. Even though many HPV vaccination programs now include females and males, for some programs HPV vaccine eligibility continues to be determined based on one's sex and only open to one sex. Moreover, providers may base HPV vaccine recommendations on individuals' sex assigned at birth, rather than their gender identity, which impacts vaccine coverage among individuals who identify as transgender [113].

The majority of the studies did not report data on participants' sexual orientation or gender identity. Lesbian, gay, bisexual, transgender and gender diverse adolescents and their perspectives are notably missing from the published studies and require due attention. The Sex and Gender Equity in Research (SAGER) guidelines were published in 2016 to provide a procedure for reporting of sex as well as gender information in study design, data analyses, results and interpretation of findings and can be applied to quantitative as well as qualitative research [114].

We note many studies lacked reflectivity with regards to how researcher's role and vaccine attitudes may have influenced participants' responses and subsequent data analysis, an observation commonly made about qualitative research studies published in healthcare journals with strict word-counts [85, 115]. It is possible participating adolescents expressed more positive views towards vaccines if they perceived study researchers were themselves supportive of vaccines. Additionally, parental consent was usually required for study enrolment. It is

therefore conceivable that adolescents from homes supportive of vaccination were more likely to participate.

## Limitations

This review only includes studies published in English language which inevitably resulted in overrepresentation of studies from Anglophone countries. We also recognize a grey literature search would provide additional and potentially richer data compared to studies published in academic journals or as graduate thesis, but this was not realistic within the scope of our work.

Studies included in the review were conducted before the introduction of COVID-19 vaccine for adolescents. Experiences with COVID-19 vaccine and the surrounding debates around adolescents' right to self-consent to the vaccine may influence adolescent views around other vaccines–a topic that warrants exploration in future studies.

We acknowledge that all authors of this systematic review are healthcare researchers with interest in addressing vaccine hesitancy. First author (HM) also does clinical work with adolescents and families, with focus on health literacy and engaging adolescents in vaccine decision-making. While we do not perceive our pro-vaccine stance or commitment to adolescent-centred care to be a limitation of our work, we do recognize that our interpretation and critique of literature findings are inevitably shaped by our professional and personal views.

## Conclusions

Our systematic review of 59 adolescent vaccine studies found adolescents often have limited understanding of vaccines and may perceive vaccine information to be directed towards their parents or caretakers rather than themselves. Because parental consent is usually required for adolescent vaccines, adolescents may not be motivated to learn about vaccines and participate in vaccine decision making. Along with parents, schools and healthcare providers have an important role in providing adolescents with appropriate vaccine information and may positively impact vaccine confidence, involvement in decision-making and potentially vaccine uptake during adolescence. Media resources are an important source of vaccine information that adolescents can access independently, but do not replace in-person conversations. Qualitative literature about adolescent vaccines would be enriched by studies examining vaccines other than the HPV vaccine, studies conducted in LMICs, and by deliberately eliciting vaccine experiences of adolescents of various gender identities and sexual orientation.

## Supporting information

**S1 Checklist. PRISMA checklist.**
(DOCX)

**S1 Text. Systematic review protocol.**
(DOCX)

**S1 Table. Search strategies.**
(DOCX)

## Acknowledgments

Helen L. Brown provided input into search strategy development. Mohammed Al-Musawi participated in the study screening process.

## Author Contributions

**Conceptualization:** Hana Mitchell, Julie A. Bettinger.

**Formal analysis:** Hana Mitchell, Rebecca Lim, Joban Dhanoa.

**Investigation:** Hana Mitchell.

**Methodology:** Hana Mitchell, Prubjot K. Gill.

**Supervision:** Ève Dubé, Julie A. Bettinger.

**Writing – original draft:** Hana Mitchell.

**Writing – review & editing:** Hana Mitchell, Rebecca Lim, Joban Dhanoa, Julie A. Bettinger.

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
