## [Decision Letter · Decision Letter 0]

21 Jul 2022

PGPH-D-22-00947

What do adolescents think about vaccines? Systematic review of qualitative studies.

Dear Dr. Bettinger,

Thank you for submitting your manuscript to PLOS Global Public Health. After careful consideration, we feel that it has merit but does not fully meet PLOS Global Public Health’s publication criteria as it currently stands. Therefore, we invite you to submit a revised version of the manuscript that addresses the points raised during the review process.

Please, follow the comments of Rewiever 1 placed below, and submit your revised manuscript by Aug 20 2022 11:59PM. If you will need more time than this to complete your revisions, please reply to this message or contact the journal office at globalpubhealth@plos.org. Please include the following items when submitting your revised manuscript:

We look forward to receiving your revised manuscript.

Kind regards,

Hanna Nalecz, Ph.D.

Academic Editor

Journal Requirements:

1. Please amend your detailed online Financial Disclosure statement. This is published with the article. It must therefore be completed in full sentences and contain the exact wording you wish to be published.

State the initials, alongside each funding source, of each author to receive each grant.

2. Please ensure that the funders and grant numbers match between the Financial Disclosure field and the Funding Information tab in your submission form. Note that the funders must be provided in the same order in both places as well.

3. Please update your online Competing Interests statement. If you have no competing interests to declare, please state: “The authors have declared that no competing interests exist.”

4. Please provide separate figure file in .tif or .eps format and ensure that all files are under our size limit of 10MB.

5. Please ensure that you refer to Table 1 in your text as, if accepted, production will need this reference to link the reader to the table.

6. We have noticed that you have uploaded Supporting Information files, but you have not included a list of legends. Please add a full list of legends for your Supporting Information files after the references list.

Additional Editor Comments (if provided):

Reviewers' comments:

Reviewer's Responses to Questions

**Comments to the Author**

1. Does this manuscript meet PLOS Global Public Health’s publication criteria? Is the manuscript technically sound, and do the data support the conclusions? The manuscript must describe methodologically and ethically rigorous research with conclusions that are appropriately drawn based on the data presented.

Reviewer #1: Yes

Reviewer #2: Yes

2. Has the statistical analysis been performed appropriately and rigorously?

Reviewer #1: N/A

Reviewer #2: N/A

3. Have the authors made all data underlying the findings in their manuscript fully available (please refer to the Data Availability Statement at the start of the manuscript PDF file)?

Reviewer #1: Yes

Reviewer #2: Yes

4. Is the manuscript presented in an intelligible fashion and written in standard English?

Reviewer #1: Yes

Reviewer #2: Yes

5. Review Comments to the Author

Reviewer #1: My general impression of this manuscript is that it is a valuable and comprehensive global review of qualitative evidence on adolescents’ understanding about vaccines, and I found it very interesting to read. This article captures useful qualitative insights with transferability across contexts. Great work! I have quite a few comments but they only request minor edits and clarifications.

Abstract

- Do you mean high-income settings (in any country) or high-income countries? If the latter, it would provide clarity to use the word countries instead of settings.

- Would “perspectives from sexual and gender minorities” or something be a better term than “homosexual and non-binary adolescents”? The latter does not include e.g. transgender adolescents who may have very specific needs or concerns around certain vaccines. It looks to me like the APA guidelines on inclusive language are quite helpful: https://www.apa.org/about/apa/equity-diversity-inclusion/language-guidelines (includes links to even more specific guides on sexual orientation and gender if relevant, please also note the caveats on the term “minorities”)

- How was the evidence synthesised? Some brief details about methods needed in abstract.

- If there is space, I think it could be useful to expand “media” to “traditional and online media” (in line with your discussion) or something else more specific as readers may assume media to only mean traditional mass media, whereas social media would be expected to be particularly relevant for adolescents.

Methods

- I am impressed that each record seems to have been double-screened, quite an undertaking!

- This section needs more details about the narrative synthesis approach and process (e.g. what data were extracted and how, how were the sub-headings of the results section developed – in advance or more inductively during the synthesis?)

Results

- I do not think “gender orientation” is typically used – for gender, the recommended terms would be “gender identity” or “gender expression” (depending on exact context), and orientation is used in the term “sexual orientation”.

- Table 1 is very useful! Just check spelling, e.g. “lead” should be “led”.

- I am very glad to see how comprehensively the individual studies are indicated in the text – it really helps to trace the evidence in the synthesis (commenting on this because not all reviews do this but it is so valuable and much more transparent!).

- For clarity, “vaccine handouts” may be better described as “vaccine information sheets” or similar (handout is used at least twice and may confuse reader to think it is about cost, not information).

- The word “vaccine” is missing when referring to the HPV vaccine on line 282 (starts with Sweden) and 283.

Discussion

- Generally very comprehensive and interesting!

- Elaborate on why inclusion and consideration of sexual and gender minorities is so important based on the sources you cite – I agree that this is key but it may not be fully obvious to every reader and the relevance of course varies somewhat depending on the vaccine and the programme/policy (e.g. HPV vaccine programmes only targeting girls would have implications for inclusion of transgender and non-binary adolescents).

Conclusion

- Again, I would recommend using “sexual and gender minorities” or at least avoiding “gender orientation” and using “identity” instead (but I think you mean both gender and sexuality here so relevant to mention both).

Reviewer #2: I have reviewed the manuscript titled “What do adolescents think about vaccines? Systematic review of qualitative studies”. In my view, the manuscript is scientifically sound (on adolescents’ views around vaccines), presents detailed methodology used, ethically describing how researchers’ background could have influenced interpretation of data collected (a strength to a qualitative research). Data underlying the findings are fully described, appropriately analyzed, available in the manuscript and supporting conclusions and recommendations made. This was a qualitative study; hence, rigorous statistical data analysis does not apply to this manuscript.

6. PLOS authors have the option to publish the peer review history of their article (what does this mean?). If published, this will include your full peer review and any attached files.

**Do you want your identity to be public for this peer review?** For information about this choice, including consent withdrawal, please see our Privacy Policy.

Reviewer #1: No

Reviewer #2: **Yes: **Switbert Rwechungura Kamazima.

---

## [Decision Letter · Decision Letter 1]

26 Aug 2022

PGPH-D-22-00947R1

What do adolescents think about vaccines? Systematic review of qualitative studies.

Dear Dr. Bettinger,

Thank you for submitting your manuscript to PLOS Global Public Health. After careful consideration, we feel that it has merit but does not fully meet PLOS Global Public Health’s publication criteria as it currently stands. Therefore, we invite you to submit a revised version of the manuscript that addresses the points raised during the review process.

We kindly recommend:

1. Specifying that you are referring to "sexual orientation" - there are several instances where the word "orientation" is used alone.

2. Elaborating in the sentence *"Only 4 studies specifically reported participants’ gender identity"* that the point you are making is that only four studies reported how participants self-identify in terms of gender, or Your point is that only four included any information about participant gender.

We look forward to receiving your revised manuscript.

Kind regards,

Hanna Nalecz, Ph.D.

Academic Editor

Journal Requirements:

1. We do not publish any copyright or trademark symbols that usually accompany proprietary names, eg (R), (C), or TM  (e.g. next to drug or reagent names). Please remove all instances of trademark/copyright symbols throughout the text, including ©, ®, ™ on page 17.

Additional Editor Comments (if provided):

Reviewers' comments:

Reviewer's Responses to Questions

**Comments to the Author**

1. If the authors have adequately addressed your comments raised in a previous round of review and you feel that this manuscript is now acceptable for publication, you may indicate that here to bypass the “Comments to the Author” section, enter your conflict of interest statement in the “Confidential to Editor” section, and submit your "Accept" recommendation.

Reviewer #1: (No Response)

2. Does this manuscript meet PLOS Global Public Health’s publication criteria? Is the manuscript technically sound, and do the data support the conclusions? The manuscript must describe methodologically and ethically rigorous research with conclusions that are appropriately drawn based on the data presented.

Reviewer #1: Yes

3. Has the statistical analysis been performed appropriately and rigorously?

Reviewer #1: N/A

4. Have the authors made all data underlying the findings in their manuscript fully available (please refer to the Data Availability Statement at the start of the manuscript PDF file)?

Reviewer #1: Yes

5. Is the manuscript presented in an intelligible fashion and written in standard English?

Reviewer #1: Yes

6. Review Comments to the Author

Reviewer #1: Thank you for addressing my comments. There are several instances where the word "orientation" alone is used without specifying that you are referring to "sexual orientation" so that still needs to be clarified throughout. I would perhaps also elaborate in the sentence "Only 4 studies specifically reported participants’ gender identity" that the point you are making is that only four studies reported how participants self-identify in terms of gender, or are you saying only four included any information about participant gender? Other than these small clarifications I am happy with the revisions.

7. PLOS authors have the option to publish the peer review history of their article (what does this mean?). If published, this will include your full peer review and any attached files.

**Do you want your identity to be public for this peer review?** For information about this choice, including consent withdrawal, please see our Privacy Policy.

Reviewer #1: No

---

## [Editor Report · Decision Letter 2]

2 Sep 2022

What do adolescents think about vaccines? Systematic review of qualitative studies.

PGPH-D-22-00947R2

Dear Dr. Bettinger,

We are pleased to inform you that your manuscript 'What do adolescents think about vaccines? Systematic review of qualitative studies.' has been provisionally accepted for publication in PLOS Global Public Health.

Best regards,

Hanna Nalecz, Ph.D.

Academic Editor